# Cancer of Unknown Primary (CUP): genetic evidence for a novel nosological entity? A case report

Silvia Benvenuti[1], Melissa Milan[1], Elena Geuna[2], Alberto Pisacane[3], Rebecca Senetta[4], Gennaro Gambardella[5,6], Giulia M Stella[7], Filippo Montemurro[2], Anna Sapino[3,4], Carla Boccaccio[8,9] & Paolo M Comoglio[1,*]

## Abstract

Cancer of unknown primary (CUP) is an obscure disease characterized by multiple metastases in the absence of a primary tumor. No consensus has been reached whether CUPs are simply generated from cancers that cannot be detected or whether they are the manifestation of a still unknown nosological entity. Here, we report the complete expression and genetic analysis of multiple synchronous metastases harvested at warm autopsy of a patient with CUP. The expression profiles were remarkably similar and astonishingly singular. The whole exome analysis yielded a high number of mutations present in all metastases (fully shared), additional mutations (partially shared) accumulated one after another in a series, and few private mutations were unique to each metastasis. Surprisingly, the phylogenetic trajectory linking CUP metastases was atypical, depicting a common "stream", sprouting a series of linear "brooks", at variance from the extensive branched evolution observed in metastases from most cancers of known origin. The distinctive genetic and evolutionary features depicted suggest that CUP is a novel nosological entity.

**Keywords** cancer of unknown primary; genetic evolution; metastasis
**Subject Categories** Cancer; Genetics, Gene Therapy & Genetic Disease

See also: **V Davalos & M Esteller** (July 2020)

## Introduction

In spite of representing 3–5% of all new cancer diagnoses, cancer of unknown primary (CUP) is the fourth most common cause of cancer-related deaths worldwide (Pavlidis & Pentheroudakis, 2012; Varadhachary & Raber, 2014) and remains a mysterious nosological entity sharing common traits: (i) early dissemination; (ii) unpredictable organ distribution; (iii) lack of tissue-specific differentiation markers; and (iv) poor prognosis. The elusive CUP biology results in the lack of effective, pathogenesis-based therapy (Golfinopoulos et al, 2009). The standard of care for CUPs is based on chemotherapy, driven by an empirical, semi-agnostic, approach based on histological suggestions from a panel of immunohistochemical markers (Fizazi et al, 2015).

Until recently, major efforts have been directed to predict the tissue of origin by means of immunohistochemistry (Greco et al, 2012), gene expression (Hainsworth et al, 2013), miRNA (Søkilde et al, 2014), or epigenetic (Moran et al, 2016) profiling, with the assumption that the knowledge of the putative tissue of origin could dictate therapeutic strategies. Yet, a recently published clinical trial showed no advantages of a molecularly defined, site-specific chemotherapy regimen compared with an empirically chosen chemotherapy (Hayashi et al, 2019). Another approach, which awaits confirmation, consists of finding druggable molecular target (s) (Ross et al, 2014). Thus, mutational profiles could be useful to reveal CUPs' vulnerabilities (the "precision medicine" approach). Genomic surveys of CUPs, performed on panels of selected cancer genes, have been recently presented (Ross et al, 2014; Löffler et al, 2016; Varghese et al, 2017; Zehir et al, 2017), but a distinguishable and specific genetic signature has not emerged and no actionable targets have been identified. Moreover, the typical multimetastatic

1  Molecular Therapeutics and Exploratory Research Laboratory, Candiolo Cancer Institute, FPO – IRCCS, Candiolo (Turin), Italy
2  Oncology Outpatient Clinic, Candiolo Cancer Institute, FPO – IRCCS, Candiolo (Turin), Italy
3  Pathology Unit, Candiolo Cancer Institute, FPO – IRCCS, Candiolo (Turin), Italy
4  Department of Medical Sciences, University of Turin, Turin, Italy
5  Telethon Institute of Genetics and Medicine (TIGEM), Pozzuoli (Naples), Italy
6  University of Naples Federico II, Naples, Italy
7  Department of Medical Sciences and Infectious Diseases, Unit of Respiratory System Diseases, IRCCS Fondazione Policlinico San Matteo, Pavia, Italy
8  Laboratory of Cancer Stem Cells, Candiolo Cancer Institute, FPO – IRCCS, Candiolo (Turin), Italy
9  Department of Oncology, University of Turin Medical School, Candiolo (Turin), Italy
   *Corresponding author. Tel: +39 0119 93601; E-mail: pcomoglio@gmail.com

presentation of CUPs might represent a further major challenge for precision medicine since the genetic makeup of each metastasis might be rather heterogeneous, undermining the outcome of therapies tailored on genetic alterations detected in the single lesion subjected to biopsy.

On the theoretical ground, it remains an open question whether CUPs are a jumble of metastatic cancers where the primary cannot be detected, or they are a still unrecognized cancer type propelled by distinctive genetic and molecular features (Pentheroudakis *et al*, 2007). To answer, we tried to decipher the evolutionary trajectories linking the multiple and synchronous metastases arising in a patient with CUP, thus providing genetic evidence for a new nosological entity and hints to envisage targeted therapeutic interventions.

# Results

## A thorough diagnosis of CUP

We studied in depth a 49-year-old male presenting with rapidly progressing multiple metastases in different sites. A thorough multistep workout was conducted following the ESMO guidelines, which excluded the presence of a primary tumor (Fizazi *et al*, 2015) (Fig 1A–D and Table EV1). Histology of an ultrasound-guided core biopsy of a breast metastasis revealed a poorly differentiated tumor with adenocarcinoma features (Fig EV1). The tumor was intensively immunoreactive for cytokeratins 7, AE1/AE3, and BCA225, and focally positive for cytokeratin 20, whereas it was negative for the markers listed in Table EV1 (Fig EV1). Cancer-associated genetic alterations scrutinized by OncoCarta™ were undetectable (Table EV2). The patient was offered a treatment in a phase 2 trial, assessing nab-paclitaxel-based doublet as first-line therapy in CUPs (AGNOSTOS trial, no. 008-IRCC-10IIS-14). Nevertheless, he progressed rapidly and after two cycles he was withdrawn from chemotherapy. Three months later the patient succumbed and underwent a "warm" autopsy. Fifteen spatially distinct metastases encompassing eight different organs/tissues were harvested: left axillary lymph node ($n = 1$), abdominal subcutis ($n = 1$), right colic flexure ($n = 1$), liver ($n = 4$), kidney ($n = 2$), gluteal subcutis ($n = 1$), mediastinum ($n = 1$), right-side breast ($n = 1$), and lung ($n = 3$). All investigated sites showed the same histology (Fig 1D), superimposable to the diagnostic breast biopsy (Fig EV1).

Cancer of unknown primary diagnosis was further confirmed at the transcriptional level. RNA-seq analysis of metastases from six sites (right colic flexure, liver, kidney, mediastinum, breast, and lung) yielded gene expression profiles that were similar among each other but did not match the profiles available in the TCGA dataset of any conventional primary tumor (i.e., tumors originated in a recognizable organ) or metastases from known primaries (Fig 2). Two primary ovarian cancers used as controls displayed expression profiles similar to that of ovarian cancers deposited in the TCGA dataset. The hierarchical clustering analysis, based on correlation distance, was performed starting from the median expression profiles of each primary cancer type or metastases calculated from data deposited in TCGA. All median expression profiles preserve the tissue-specific identity feature (i.e., each metastasis clusters close to its tumor of origin; Fig EV2B). The transcriptional profiles of CUP metastases were unrelated to any putative tissue of origin (unlike

metastases originated from known primaries) and enlightened a distinct expression signature.

## Genomic characterization

The genomic DNA extracted from the fifteen metastases was analyzed by whole exome sequencing (WES) and compared with the patient's own peripheral blood mononuclear cell DNA. The average depth of coverage was about 100× (Fig EV3). Single nucleotide variants (SNVs) and small insertions and deletions (InDels) were called with Strelka2 (Kim *et al*, 2018) to identify somatic alterations (Dataset EV1).

The presence of germline mutations in the eight genes responsible for the main hereditary human tumors (BRCA1, BRCA2, CDH1, CTNN1A, MLH1, MSH2, MSH6, and p53), suspected because of the patient's familial history of multiple cancers, was excluded.

In the fifteen CUP metastases, the genetic analysis yielded a number of nonsynonymous SNVs/InDels ranging from 144 to 376 (Fig 3A) for a total of 748 unique changes. Mutation rate varied from 6.00 to 8.00 mutations per $10^6$ bases, thus excluding that any metastases were hyper-mutated (Network, 2012). The predominant signature of C>A transversions observed was consistent with a smoking signature (Alexandrov *et al*, 2016). All samples were microsatellite stable (Boland & Goel, 2010).

The mutations identified were classified as (i) fully shared, (ii) partially shared, or (iii) private. (i) We considered fully shared the mutations displayed by at least 80% of metastases (i.e., 12/15). This threshold was chosen since differences in coverage, at any given region, would potentially produce false positives (i.e., mutations present only in one or few metastases) or false negatives (i.e., the absence of a mutation in one or more samples, due to insufficient coverage). By these criteria, 276 mutations were fully shared. (ii) On top of this common mutational pattern, additional mutations accumulated incrementally in each metastasis. These mutations, added one after the other in different metastases, were defined as partially shared. (iii) Finally, a few private mutations (from 4 to 48) were unique to each metastasis (Fig 3A). Indeed, the genetic concordance of the fifteen metastases, measured as a function of Jaccard similarity, ascertains a low degree of inter-metastases heterogeneity as all metastasis (but lesion L_8) display a similarity ranging from 58 to 82% (Figs 3B and EV4). The exception of L_08 is due to the scarce number of mutations displayed; nevertheless, out of the 144 somatic SNVs observed, four were partially shared and only four private. All remaining mutations were fully shared with other metastases.

The clonal composition of the fifteen metastases, performed by clustering the variant allele frequency according to the algorithms SciClone (Miller *et al*, 2014) and ClonEvol (Dang *et al*, 2017), varied from 1 to 8, mostly following a linear pattern of evolution (Fig 3C). Clones harboring the same mutations expanded at different rates in different metastases (Fig 3D).

Although identification of the genes involved in CUP onset and progression is not the focus of this analysis, it is worth to mention a few mutations occurring in known tumor-associated genes. In fact, within the 276 fully shared mutations (Dataset EV1), we found genetic lesions in the oncogene NTRK1 and the tumor suppressors TP53, ARID2, SMARCA4, ZFHX3, all of which have been described in CUPs (Zehir *et al*, 2017).

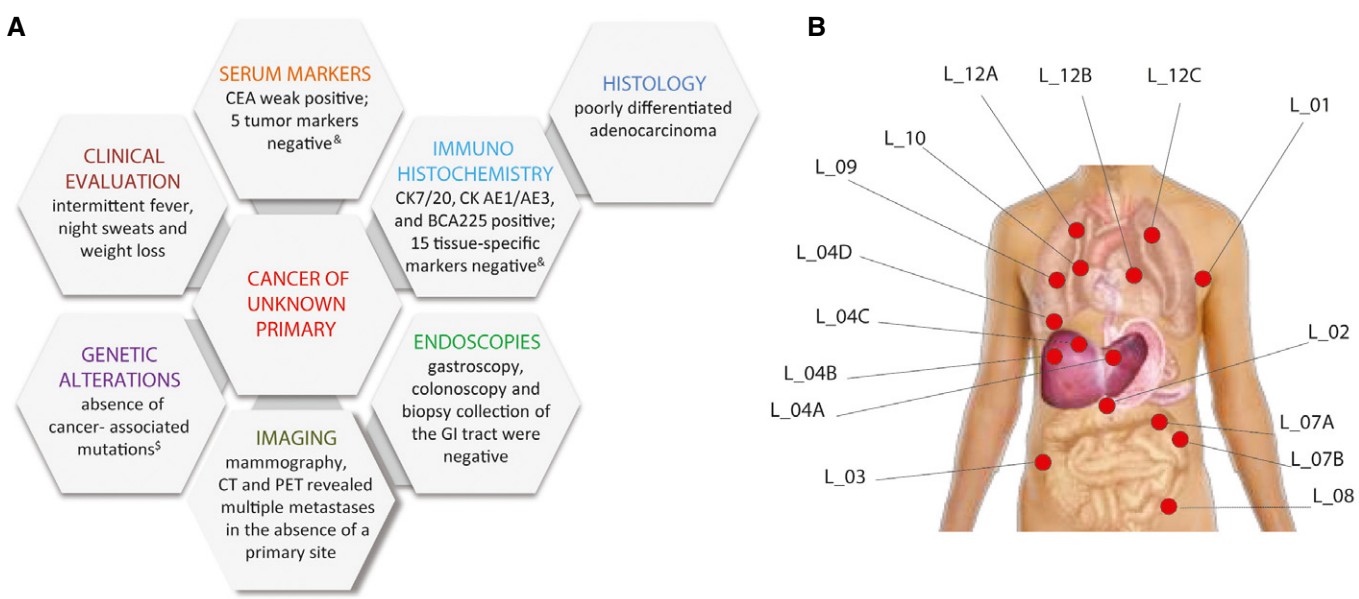

**C**

| Metastasis ID | Anatomic site |
|---|---|
| L_01 | Left axillary lymph node |
| L_02 | Abdominal subcutis |
| L_03 | Right colic flexure |
| L_04A | Liver lesion A |
| L_04B | Liver lesion  B |
| L_04C | Liver lesion  C |
| L_04D | Liver lesion  D |
| L_07A | Kidney lesion A |
| L_07B | Kidney lesion B |
| L_08 | Gluteal subcutis |
| L_09 | Mediastinum |
| L_10 | Right sided breast |
| L_12A | Lung lesion A |
| L_12B | Lung lesion B |
| L_12C | Lung lesion C |

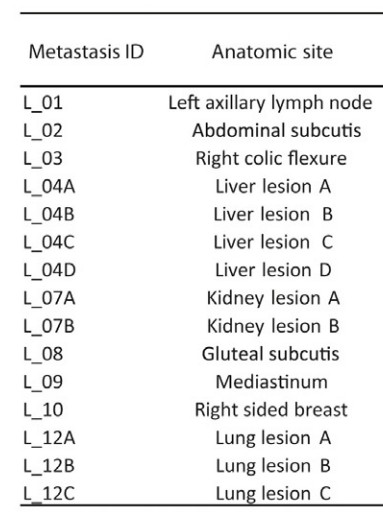

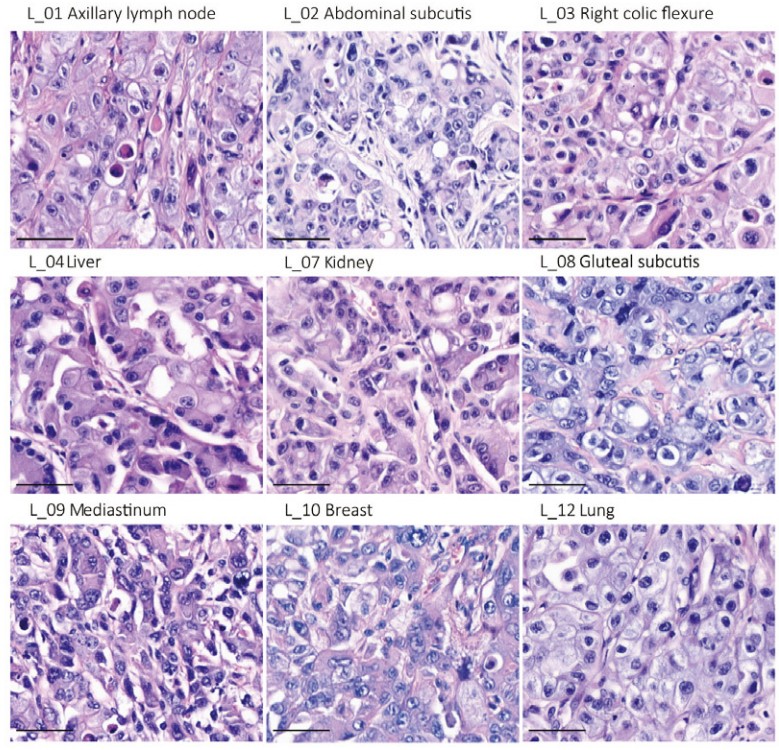

**Figure 1. CUP diagnosis.**

A  Multistep *ad excludendum* diagnostic workflow: diagnosis performed in accordance with ESMO guidelines (Fizazi *et al*, 2015), starting from clinical evaluation and proceeding with the sequential examinations represented in clockwise order. &Serum and immunohistochemistry markers are listed in Table EV1. $Cancer-associated genes are listed in Table EV2. GI: gastrointestinal.

B, C  Metastases distribution: The fifteen metastases were retrieved at warm autopsy from 8 tissues/organs. Samples are numbered according to the sequence of harvest at autopsy.

D  Histology: All metastatic lesions are composed of poorly differentiated cells with an epithelial "flavor". The neoplastic population is mainly arranged in solid nests and sheets with focal rudimental gland formation. Scale bar: 50 μm.

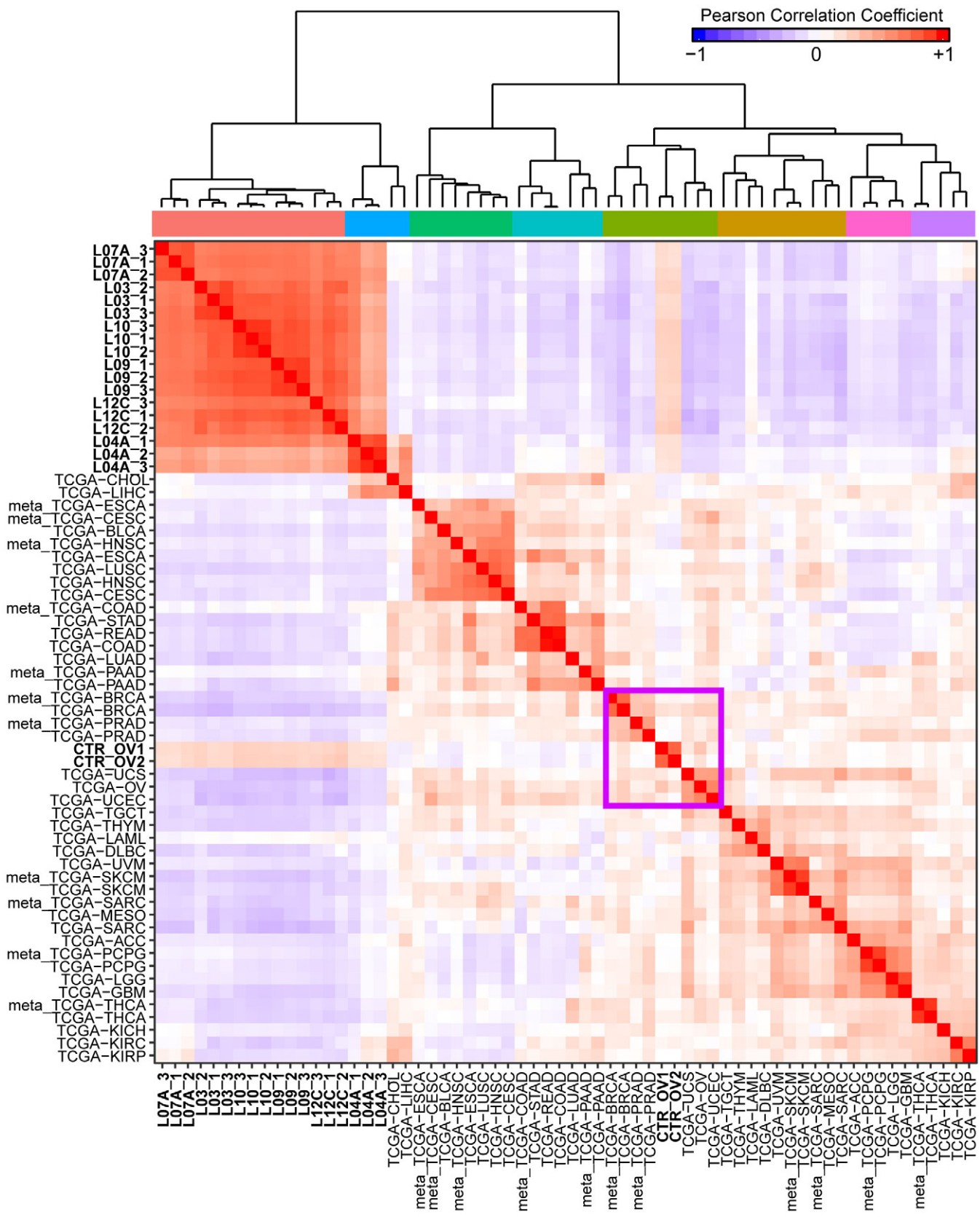

**Figure 2.**

**Figure 2.   Hierarchical clustering analysis of gene expression profiles of six CUP metastases.**
Triplicate samples of L_03, L_04A, L_07A, L_09, L_10, and L_12C are compared with the expression profiles deposited in the TCGA dataset of a spectrum of primary tumors or metastases (meta) from known origin. Two ovarian cancers analyzed in house (CTR_OV1 and CTR_OV2) were used as controls and have expression profiles matching the profiles displayed by the ovarian cancers listed in TCGA (purple box). The acronyms are as follow: ACC, adrenocortical carcinoma; BLCA, bladder urothelial carcinoma; BRCA, breast invasive carcinoma; CESC, cervical squamous cell carcinoma and endocervical adenocarcinoma; CHOL, cholangiocarcinoma; COAD, colon adenocarcinoma; DLBC, lymphoid neoplasm diffuse large B-cell lymphoma; ESCA, esophageal carcinoma; GBM, glioblastoma multiforme; HNSC, head and neck squamous cell carcinoma; KICH, kidney chromophobe; KIRC, kidney renal clear cell carcinoma; KIRP, kidney renal papillary cell carcinoma; LAML, acute myeloid leukemia; LGG, brain lower grade glioma; LIHC, liver hepatocellular carcinoma; LUAD, lung adenocarcinoma; LUSC, lung squamous cell carcinoma; MESO, mesothelioma; OV, ovarian serous cystadenocarcinoma; PAAD, pancreatic adenocarcinoma; PCPG, pheochromocytoma and paraganglioma; PRAD, prostate adenocarcinoma; READ, rectum adenocarcinoma; SARC, sarcoma; SKCM, skin cutaneous melanoma; STAD, stomach adenocarcinoma; TGCT, testicular germ cell tumors; THCA, thyroid carcinoma; THYM, thymoma; UCEC, uterine corpus endometrial carcinoma; UCS, uterine carcinosarcoma; UVM, uveal melanoma.

### Phylogenetic tree reconstruction

Exploiting the whole exome analysis of the multiple metastases harvested from the same patient, and taking into account both synonymous and nonsynonymous mutations (SNVs and InDels) occurring in each metastasis and their purity-corrected frequencies (Phylip tool; Felsenstein, 2005), we reconstructed the phylogenetic relationships. While previous studies on metastases disseminated by known primary tumors revealed branched patterns mostly modeled as trees (Gerlinger *et al*, 2012) or stars (Sottoriva *et al*, 2015), analysis of the patient with CUP identified a single common "stream", sprouting a series of sequential linear "brooks". Figure 3E shows the sequential alignment of the fifteen metastases, based on the phylogenetic tree inferred by Phylip implemented by private mutations observed in each metastasis (Fig EV5). This picture suggests an unusual phylogenetic evolution consistent with the existence of a common ancestor that continues to accumulate mutations in a linear fashion, releasing over time collateral branches (the different metastases), each of which accrues an independent smaller mutational burden. The common ancestor is obviously undetectable in the patient body and may not necessarily display the features of a conventional cancer stem cell that generates a primary tumor mass. Rather, this ancestor might release in the bud its evolving progeny, which would rapidly disseminate and form metastases in tissues where microenvironmental conditions favor settlement and local growth.

## Discussion

The conventional approach to therapy of cancers of unknown primary relies in pushing the molecular characterization of the metastatic lesions to the limits, to bet on a putative tissue of origin, and to treat the patients as if they were affected by a highly metastatic cancer of that tissue (Hayashi *et al*, 2019). It is possible that by this approach more (or possibly all) currently defined CUPs will be re-classified in a "tissue-gnostic" way. As an alternative, CUPs might turn out to be a nosological entity with distinctive traits. Identification of the gene(s) and the molecules responsible for these traits could provide hints to understand the biology of early tumor dissemination, from one side, and to pinpoint new selective therapeutic targets from the other. We approached the problem by a comprehensive comparative analysis of the transcriptional profiles, the genetic traits, and the phylogenetic relationship among multiple synchronous and spatially distinct metastases in an exemplary case of CUP. Such information has never been reported.

The first surprise was the expression profiles shared by CUP metastases which were, otherwise, dissimilar from the profiles displayed by the plethora of normal and tumor cells (including metastases) deposited in the TCGA dataset.

The second unexpected finding was the high degree of similarity among the mutational makeup of different CUP metastases, unlike what commonly observed among metastases from tumors of known origin (Gerlinger *et al*, 2012; Sottoriva *et al*, 2015). This similarity is surprising since the precocious dissemination of the disease would suggest high inter-metastases heterogeneity according to the parallel progression model (Naxerova & Jain, 2015). Indeed, when metastases from known primary tumors disseminate early, they continue to evolve independently, giving rise to a wide genetic divergence. However, the high degree of homogeneity among CUP metastases is consistent with the rapid clinical evolution: after homing into multiple tissues, founder cells generate metastases leading patients to death in such short a time that only minimal divergent evolution can take place.

The incremental accumulation of "partially shared" and the presence of few "private" mutations in individual metastases allowed drawing the phylogenetic tree. The inferred evolutionary pattern is unusual in metastases originated from known primary tumors. As described, the phylogenetic trajectory depicted a "stream-like" path from which a number of linear "brooks" originated. This pattern is surprisingly reminiscent of the expansion of a galaxy (Fig EV5). An alternative botanical metaphor recalls the olive tree terminal shoot.

This model is consistent with the presence of a common cell of origin with stem-like features that after accumulating the common set of mutations, including all those affecting the putative driver genes, became fully malignant, and acquired the ability to relentlessly proliferate and disseminate its progeny. Such progeny likely underwent further accrual of the "partially shared" mutations and modest divergence, in the meantime spreading across the organism. At different metastatic sites, the founder cell(s) generated metastases where the accumulation of "private" mutations was minimal.

Although the identification of possible metastatic drivers is beyond the scope of the paper, the mutational analysis enlightened a few candidate genes—shared by all metastases—and many possible gene combinations that may interfere with key signaling pathways controlling invasive growth (Comoglio *et al*, 2018). Among the mutations conserved in all metastatic sites, it is worth mentioning truncation of TP53, and critical amino acid substitutions in the transcription factor ZFHX3, in the receptor NTRK1, and in the chromatin remodeling proteins ARID2 and SMARCA4, already suspected to be implicated in CUP pathogenesis (Zehir *et al*, 2017).

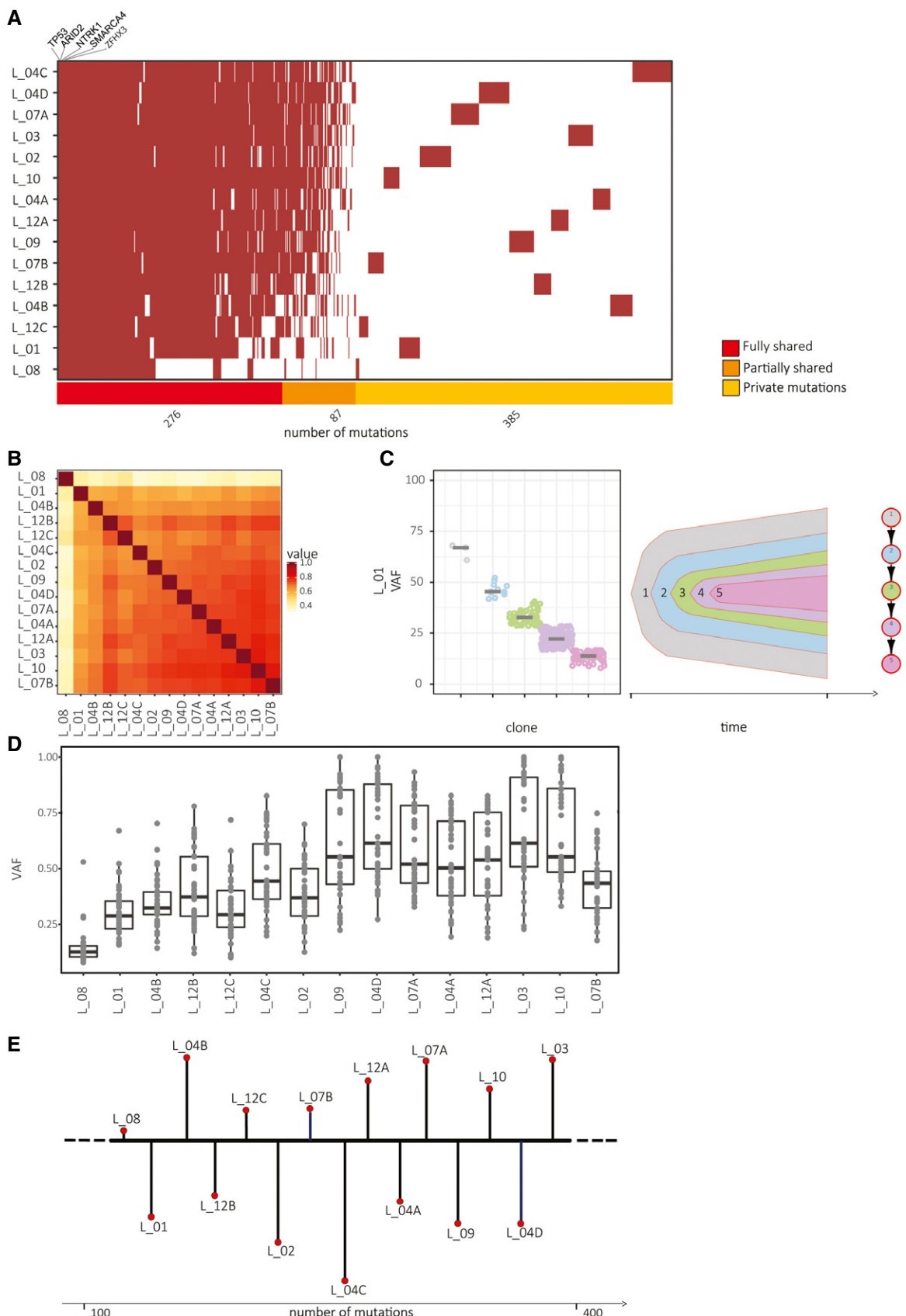

Figure 3.

◀

**Figure 3. Genetic analysis of fifteen CUP metastases.**

A Somatic mutations distribution: SNVs and InDels displayed by each metastasis (listed in the *y*-axis) were detected by Strelka2 tool (Kim *et al*, 2018). Brown traits represent single mutations. The number of fully shared (red), partially shared (orange), or private (yellow) mutations is indicated in the horizontal bar below. Five cancer-associated genes (TP53, ARID2, NTRK1, SMARCA4, and ZFHX3) mutated in all metastases are highlighted.

B Genetic similarity among metastases: the heatmap has been drawn according to the Jaccard index. The similarity ranges from 58 to 82% among all pairs with the exception of L_08 due to the scarce number of mutations.

C Clonal composition analysis: representative nested view of metastasis L_01. The five sub-clones (represented in different colors) inferred by clustering similarities of the variant allele frequencies (VAFs) follow a linear pattern.

D Variant allele frequency of fully shared mutations present in copy neutral regions. The box plot represents the fluctuation of the VAF (in the *y*-axis) of single mutations (grey dots) in each metastasis; each box represents the upper and lower quartiles, while the central short black line within each box represents the median; whiskers indicate variability outside the upper and lower quartiles.

E Reconstruction of the phylogenetic tree linking the fifteen CUP metastases: the model was reconstructed by taking into account the maximum likelihood molecular evolutionary tree according to Phylip (Felsenstein, 2005) and the incremental number of mutations. "Brooks" lengths are proportional to the amount of "private" mutations.

Data gathered in this paradigmatic patient suggest that cancers of unknown primary behave as a distinct nosological entity. Although collection of multiple samples from a single patient is not trivial, further accrual of cases is required to strength the hypothesis. On the clinical ground, the unexpected genetic similarity among different CUP metastases leaves room for a therapeutic strategy aimed at the simultaneous eradication of multiple lesions.

## Materials and methods

### Patient recruitment, diagnosis, and tissue collection

Patient was enrolled at Candiolo Cancer Institute within AGNOSTOS Trial (no. 008-IRCC-10IIS-14) approved by the Institute Ethical Committee. Informed consensus was obtained from patient, and the experiments were conformed to the principles set out in the WMA Declaration of Helsinki and the Department of Health and Human Services Belmont Report. CUP diagnosis was made following the *ad excludendum* diagnostic workflow in accordance with ESMO guidelines (Fizazi *et al*, 2015). Fresh human specimens were collected during a "warm" autopsy and either stored in RNAlater (Life Technologies) or fixed in 4% buffered formaldehyde and embedded in paraffin.

### Immunohistochemistry and fluorescence *in situ* hybridization

Sections were either stained with hematoxylin and eosin or collected on Superfrost plus slides and used for immunohistochemistry (IHC) and fluorescence *in situ* hybridization (FISH) analyses. IHC was performed using the antibodies listed in Table EV3 and revealed with Liquid DAB + Substrate Chromogen System (K3468; Dako) using Ventana Benchmark ultra System (Roche), Bond Max (Leica Biosystems), or Autostainer Link 46 (Agilent). FISH was carried out using the Histology FISH Accessory kit (DAKO) and the probes listed in Table EV3. Images were acquired using an Olympus BX61 microscope (Olympus Corporation) and analyzed using CytoVison software (Leica Biosystems).

### RNA extraction, libraries preparation, and sequencing

Samples were macrodissected to select tumor cells before RNA extraction; after macrodissection percent of tumor cells—assessed independently by two pathologists—was above 70% in every single specimen. Total RNA was extracted from three different regions of each metastases retrieved at warm autopsy and stored in RNAlater solution by Maxwell® RSC Instrument (Promega) using Maxwell® RSC miRNA Tissue Kit (Promega). Quantification was performed on a Bioanalyzer 2100 (Agilent) using RNA 6000 nano Kit (Agilent). RNA-seq analysis was performed on six metastases (right colic flexure, liver, kidney, mediastinum, breast, and lung), and the choice was dictated by quality controls. Libraries were prepared with Illumina TruSeq Stranded mRNA kit starting from 600 ng of total RNA, and samples were fragmented and amplified for 15 PCR cycles. Libraries were size selected with Blue Pippin (Sage Science) using 1.5% gel cassettes and 350–550 bp regions isolated. Sequencing was performed in 75 paired ends with NextSeq 500 (Illumina) using NextSeq 500/550 High Output kit v2 (150 cycles).

### Sequence alignment and expression profiles

Each FASTQ file was aligned using HISAT v. 2.1.0 (Kim *et al*, 2015) using hg19 as genome reference. Transcripts assembly was performed with StringTie v. 1.3.33 and quantification performed using gffcompare v. 0.10.1 (https://ccb.jhu.edu/software/stringtie/gffcompare.shtml). The estimated abundance for the transcripts was expressed as FPKM values (Fragments Per Kilobase of transcript per Million mapped reads) (Pertea *et al*, 2016). The data were transformed into gene-level quantification by summing the FPKM of the transcripts associated with the same gene and transformed into $\log_2(\text{FPKM} + 1)$. To compare the expression profiles of eight CUP samples and two controls (ovarian cancers) with those deposited in the TCGA dataset, a median expression profile of 33 tumor types was created. TCGA expression data profiles were retrieved as FPKM using the TCGAbiolinks package (Colaprico *et al*, 2016). The analysis was limited to samples labeled as "Primary Solid Tumor", "Primary Blood Derived Cancer—Peripheral Blood", or "Primary Blood Derived Cancer—Bone Marrow". For each gene and each of the 33 tumor types, we extracted the median FPKM across all samples to generate the median expression profile of each tumor type and the profiles transformed into $\log_2(\text{FPKM} + 1)$. TCGA transcriptional profiles were normalized together using the normalize quantiles function of preprocessCore package in the R statistical environment v3.6 (Fig EV2A). Clustering analysis was performed using hclust function of R statistical environment v3.6 and ward.D2 as agglomeration method. The

**The paper explained**

**Problem**
Cancer of unknown primary is an obscure disease characterized by multiple metastases in the absence of a clinically detectable primary tumor. The elusive CUP biology results in the lack of pathogenesis-based therapy.

**Results**
Fifteen synchronous metastases from a single CUP patient were analyzed by whole exome and RNA sequencing and their phylogenetic tree was reconstructed. Surprisingly, a high percentage of mutations, including those in putative driver genes, were fully shared. Additional mutations accumulated one after the other in a series, and a few private mutations were unique to each metastasis. The phylogenetic trajectory linking CUP metastases depicted an evolution pattern reminiscent of a galaxy: a common "stream" sprouting a series of linear "brooks".

**Impact**
The distinctive genetic and evolutionary features of CUPs suggest a biology different from metastases of cancers of known origin. On the clinical ground, the unexpected genetic similarity among different CUP metastases leaves room for a therapeutic strategy aimed at the simultaneous eradication of multiple lesions.

distances among transcriptional profiles were computed as one minus the person correlation coefficient while clusters were identified using dynamicTreeCup package (Langfelder *et al*, 2008) in the R statistical environment v3.6.

### gDNA extraction, library preparation, and sequencing

gDNA was isolated using Relia Prep™ gDNA Tissue Miniprep System (Promega). Normal gDNA was derived from peripheral blood mononuclear cells (PBMCs) of the same patient using Relia-Prep™ Blood gDNA Miniprep System (Promega). DNA was quantified using Nanodrop ND1000 spectrophotometer (Thermo Fisher Scientific) and Qubit 4 Fluorometer (Thermo Fisher Scientific).

Whole exome sequencing with 150-bp paired reads was performed with a NextSeq 500 (Illumina), using 1 μg genomic DNA and enrichment for whole exome according to SeqCap EZ MedExome (Roche).

### Sequence alignment and variant annotation

Adapters were clipped using Scythe (https://github.com/vsbuffalo/scythe) and 3′ ends with a quality score < 20 over a window of 10 bases were trimmed using Sickle (Joshi & Fass, 2011), entirely removing the fragment if the final length of one of the reads was lower than 50 bp. Sequencing reads from each sample were aligned to the human genome (hg38) using Burrows–Wheeler Aligner (BWA) mem (Li & Durbin, 2010) with default parameters. PCR duplicates were removed using rmdup of SAMtools (Li *et al*, 2009). Only reads uniquely mapping in the targeted regions were considered and retained for further analysis. Somatic SNVs and small insertion/deletions (InDels) were identified using Strelka2 (Kim *et al*, 2018). Somatic SNVs and InDels were further retained if (i) supported by at least 10 mutated reads in the tumor, (ii) had allele frequency ≥ 5%, (iii) supported by less than one mutated reads in the normal, and (iv) had a reported Empirical Variant Scoring (EVS) by Strelka2 ≥ 15. ANNOVAR (Wang *et al*, 2010) was used to annotate nonsilent (nonsynonymous, stopgain, stoploss, frameshift, nonframeshift, and splicing modifications) somatic mutations in each tumor.

### Microsatellite stability analysis

Microsatellite instability was analyzed with the MSI Analysis System kit, Version 1.2 (Promega). Samples displaying variation of at least two markers are considered instable.

### Clonal evolution

The clonal structure of each metastasis was inferred with SciClone (Miller *et al*, 2014), with default parameters with the exception of minDepth that was set equal to 75. As input, all the somatic mutations (including the synonymous) were used. Copy number regions were identified by CopywriteR package (Kuilman *et al*, 2015) in order to exclude from the analysis SNVs falling in copy number altered regions of the genome. Phylogeny of each metastasis was inferred using the ClonEvol R package (Dang *et al*, 2017) with default parameters using as input the cluster of mutations identified by SciClone.

### Phylogenetic tree reconstruction among metastasis

Phylip (Felsenstein, 2005) (maximum likelihood-based method) was used to reconstruct the phylogeny among the multiple metastases of the patient with CUP. As input, we used the same mutations previously considered for SciClone. The trees in Newick format produced by Phylip were finally rendered using the R package APE (Paradis *et al*, 2004).

## Data availability

WES and RNA-seq data have been deposited in the EGA (European Genome-Phenome Archive) with the accession number EGAS00001004059 (https://ega-archive.org/studies/EGAS00001004059).

**Expanded View** for this article is available online.

### Acknowledgements

We thank V. Nigro and D. Cacchiarelli for discussing the bioinformatic data; E. Berrino and T. Venesio for MSS analysis; I. Sarotto for Immunohistochemical tests; L. Casorzo and M. Panero for FISH tests; and A. Balsamo for clinical data management. The invaluable secretarial help of A. Cignetto is acknowledged. The results on the expression profiles are in part based upon data generated by the TCGA Research Network: https://www.cancer.gov/tcga. The research leading to these results has received funding from FONDAZIONE AIRC under 5 per Mille 2018—ID. 21052 program—PI: P. Comoglio; GLs: C. Boccaccio, F. Montemurro, and A. Sapino; Italian Ministry of Health Ricerca Corrente 2019; and FPRC 5xmille 2014 Ministero Salute. GG was supported by the STAR (Sostegno Territoriale alle Attività di Ricerca) Grant of University of Naples Federico II.

## Author contributions

Conceptualization: SB, CB, EG, and PMC; Formal analysis: GG and MM; Funding acquisition: CB, PMC, and AS; Investigation: SB and MM; Resources: EG, FM, AP, AS, RS, and GMS; Supervision: PMC; Visualization: SB, GG, and MM; Writing: SB, CB, EG, and PMC.

## Conflict of interest

The authors declare that they have no conflict of interest.

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
