## [Review Process File · EMBO Molecular Medicine]

Cancer of Unknown Primary (CUP): genetic evidence for a novel nosological entity? A case report.

Silvia Benvenuti, Melissa Milan, Elena Geuna, Alberto Pisacane, Rebecca Senetta, Gennaro Gambardella, Giulia Stella, Filippo Montemurro, Anna Sapino, Carla Boccaccio, and Paolo Comoglio
DOI: [10.15252/emmm.201911756](https://doi.org/10.15252/emmm.201911756)

Corresponding author(s): Paolo Comoglio (pcomoglio@gmail.com)

Review Timeline:

Submission Date:	28th Nov 19
Editorial Decision:	16th Jan 20
Revision Received:	10th Apr 20
Editorial Decision:	4th May 20
Revision Received:	8th May 20
Accepted:	12th May 20

Editor: Lise Roth

Transaction Report:

16th Jan 2020

Dear Prof. Comoglio,

Thank you for the submission of your manuscript to EMBO Molecular Medicine. We have now received feedback from two of the three reviewers who initially agreed to evaluate your manuscript. Given that referee #2 has unfortunately not returned his/her report so far despite several chasers, and that both referees 1 and 3 are overall positive, we prefer to make a decision now in order to avoid further delay in the process. Should referee #2 provide a report, we will send it to you, with the understanding that we would not ask you for extensive experiments in addition to the ones required in the enclosed reports.

As you will see from the reports below, while reviewer #3 is overall supportive of the manuscript pending minor revisions, reviewer #1 recognizes the value of your data, but also raises a number of concerns. In particular, more samples should be analysed (normal samples from CUP patients, metastasis samples from warm autopsies, human stem cells), and clustering analysis should be performed in addition to the UMAP analysis.

Addressing the other reviewers' concerns will be necessary for further considering the manuscript in our journal. EMBO Molecular Medicine encourages a single round of revision only and therefore, acceptance or rejection of the manuscript will depend on the completeness of your responses included in the next, final version of the manuscript. For this reason, and to save you from any frustrations in the end, I would strongly advise against returning an incomplete revision. Should you find that the requested revisions are not feasible within the constraints outlined here and prefer, therefore, to submit your paper elsewhere, we would welcome a message to this effect.

When submitting your revised manuscript, please carefully review the instructions that follow below. Failure to include requested items will delay the evaluation of your revision:

- 1) A .docx formatted version of the manuscript text (including legends for main figures, EV figures and tables). Please make sure that the changes are highlighted to be clearly visible.
- 2) Individual production quality figure files as .eps, .tif, .jpg (one file per figure).
- 3) A .docx formatted letter INCLUDING the reviewers' reports and your detailed point-by-point responses to their comments. As part of the EMBO Press transparent editorial process, the point-by-point response is part of the Review Process File (RPF), which will be published alongside your paper.
- 4) A complete author checklist, which you can download from our author guidelines (<https://www.embopress.org/page/journal/17574684/authorguide#submissionofrevisions>). Please insert information in the checklist that is also reflected in the manuscript. The completed author checklist will also be part of the RPF.
- 5) Please note that all corresponding authors are required to supply an ORCID ID for their name

upon submission of a revised manuscript.

6) Before submitting your revision, primary datasets produced in this study need to be deposited in an appropriate public database (see <https://www.embopress.org/page/journal/17574684/authorguide#dataavailability>). Please remember to provide a reviewer password if the datasets are not yet public. The accession numbers and database should be listed in a formal "Data Availability" section (placed after Materials & Method). Please note that the Data Availability Section is restricted to new primary data that are part of this study.

7) We would also encourage you to include the source data for figure panels that show essential data. Numerical data should be provided as individual .xls or .csv files (including a tab describing the data). For blots or microscopy, uncropped images should be submitted (using a zip archive if multiple images need to be supplied for one panel). Additional information on source data and instruction on how to label the files are available at .

8) Our journal encourages inclusion of *data citations in the reference list* to directly cite datasets that were re-used and obtained from public databases. Data citations in the article text are distinct from normal bibliographical citations and should directly link to the database records from which the data can be accessed. In the main text, data citations are formatted as follows: "Data ref: Smith et al, 2001" or "Data ref: NCBI Sequence Read Archive PRJNA342805, 2017". In the Reference list, data citations must be labeled with "[DATASET]". A data reference must provide the database name, accession number/identifiers and a resolvable link to the landing page from which the data can be accessed at the end of the reference. Further instructions are available at .

9) We replaced Supplementary Information with Expanded View (EV) Figures and Tables that are collapsible/expandable online. A maximum of 5 EV Figures can be typeset. EV Figures should be cited as 'Figure EV1, Figure EV2' etc... in the text and their respective legends should be included in the main text after the legends of regular figures.

- Additional Tables/Datasets should be labeled and referred to as Table EV1, Dataset EV1, etc. Legends have to be provided in a separate tab in case of .xls files. Alternatively, the legend can be supplied as a separate text file (README) and zipped together with the Table/Dataset file. See detailed instructions here: .

10) For more information: There is space at the end of each article to list relevant web links for further consultation by our readers. Could you identify some relevant ones and provide such information as well? Some examples are patient associations, relevant databases, OMIM/proteins/genes links, author's websites, etc...

11) Every published paper now includes a 'Synopsis' to further enhance discoverability. Synopses

are displayed on the journal webpage and are freely accessible to all readers. They include a short stand first (maximum of 300 characters, including space) as well as 2-5 one-sentences bullet points that summarizes the paper. Please write the bullet points to summarize the key NEW findings. They should be designed to be complementary to the abstract - i.e. not repeat the same text. We encourage inclusion of key acronyms and quantitative information (maximum of 30 words / bullet point). Please use the passive voice. Please attach these in a separate file or send them by email, we will incorporate them accordingly.

Please also suggest a striking image or visual abstract to illustrate your article. If you do please provide a jpeg file 550 px-wide x 400-px high.

12) As part of the EMBO Publications transparent editorial process initiative (see our Editorial at <http://embomolmed.embopress.org/content/2/9/329>), EMBO Molecular Medicine will publish online a Review Process File (RPF) to accompany accepted manuscripts.

In the event of acceptance, this file will be published in conjunction with your paper and will include the anonymous referee reports, your point-by-point response and all pertinent correspondence relating to the manuscript. Let us know whether you agree with the publication of the RPF and as here, if you want to remove or not any figures from it prior to publication.

I look forward to receiving your revised manuscript.

Yours sincerely,

Lise Roth

Lise Roth, PhD
Editor
EMBO Molecular Medicine

To submit your manuscript, please follow this link:

Link Not Available

***** Reviewer's comments *****

Referee #1 (Remarks for Author):

Cancer of Unknown Primary (CUP): genetic evidence for a novel pathological entity.
Benvenuti et al.

Authors performed a genomic and transcriptomic characterization of several metastases of a case of cancer of unknown primary (CUP). A total of 15 synchronous metastases were retrieved at warm autopsy from 9 different tissues (liver, kidney, breast, lung, axillary lymph node, abdominal subcutaneous tissue, mediastinum, gluteal subcutis and right colic flexure). Whole exome sequencing was performed for the 15 metastases; and RNA sequencing was achieved for 6 metastases (in triplicates).

Comprehensive genomic and transcriptomic data generated in this study provide a valuable raw material to improve our understanding about the mechanistic origin and evolution of CUPs. However, a more exhaustive analysis should be performed to support author's conclusions.

Main comments:

1. Transcriptomic analysis (RNAseq): It is extremely surprising that CUP samples segregate completely apart from any other human tissue (TCGA normal, primary and metastatic samples from 33 tissue types). Although a couple of primary ovarian cancer from the group were included to exclude batch effects, additional samples should be included in the analysis to strongly support a distinct expression signature of the studied CUP samples, and robustly discard artefacts. Please include normal samples from the CUP patient (at least normal liver, kidney, breast and lung) obtained exactly as the analysed CUP samples (warm autopsy). Are they segregating in the expected TCGA normal tissue? Include additional metastasis samples obtained from warm autopsies (if possible, from your same institution). Choose representative individual metastatic samples from TCGA and plot them in Fig. 2. Is the median expression profile preserving the tissue-type identity features?

Considering that authors suggest that their model is consistent with the presence of a common cell of origin with stem-like features... and CUP features suggest a common genetic mechanism that fixes the cells in a "stem" state... it could be interesting to include human stem cells (embryonic and adult stem cells from the literature) in the transcriptomic analysis and plot them in Fig. 2. Are CUP samples showing stem-like traits instead of differentiated-like expression profiles?

In addition, gene set enrichment and pathway analysis could be performed using RNA seq data from CUP samples to assess the activation of stem-associated pathways. Moreover, expression of stem-cell markers could be shown. The idea of a common cell of origin with stem-like features is interesting, but some data supporting this hypothesis should be shown. Is the "poorly differentiated" histology consistent with the stem-like traits? Are stem-cell markers expressed in those tissues (IHC)?

Please include methodological and experimental details: Did you performed macrodissection to select tumor cells before RNA extraction? How do you calculate the median expression profile for the TCGA samples?

In addition to the UMAP analysis, please include a clustering analysis. Although UMAP analysis is useful for data visualization and could be used as an efficient preprocessing step, UMAP does not preserve density and linearity. Thus, it is strongly suggested to perform clustering analysis (hierarchical clustering and/or k-means clustering). Please, as previously recommended, include

normal tissue samples from CUP patient, other autopsy samples, and stem-cells.

2. Genomic analysis (Whole exome seq) and phylogenetic tree reconstruction: One of the most interesting finding of the study is the identification of a single common "stream", sprouting a series of sequential linear "brooks"... suggesting an unusual phylogenetic evolution consistent with the existence of a common ancestor that continues to evolve in a linear fashion...

In order to assess the significance of this data, it is necessary to evaluate the context. Thus, it could be useful to include an external "outlier" to measure the significance of the changes.

Once revised the transcriptomic and genomic analyses, please rethink the manuscript title, more considering that only one case has been analysed. Although analysis of 15 synchronous metastases from a CUP case is per se very interesting and could improve our understanding about the mechanistic origin and evolution of CUPs and shed lights about the reasons of unusual patterns of dissemination of CUPs, do not forget that is only one case. Before considering CUP as a novel pathological entity & distinct nosological entity based on your genomic analysis, additional cases should be analysed in future studies.

Minor comments:

1. Revise the Fig. 3C graph & legend. L_10 or L_01?
2. Please, do not overuse quotes (" ") and avoid terms as "flavour".
3. Include significance of abbreviation in Fig EV1 (onT).

Referee #3 (Remarks for Author):

This paper describes the genetic characteristics of several metastatic lesions from a single patient affected by a cancer of unknown origin (CUP), with the aim of getting an insight into the clonal evolution of the disease.

The authors show that the genetic evolution of this case of CUP is "consistent with the existence of a common ancestor that continues to evolve in a linear fashion, releasing over time collateral branches (the different metastases), each of which accrues an independent smaller mutational burden". This is in clear contrast with what we generally know about metastatic cancers.

The materials and methods are appropriate, especially with regard to the selection of the clinical case. The results are clearly explained and compelling.

The findings reported by the authors are quite stunning and could have a significant therapeutic impact. In particular, these results corroborate the hypothesis that CUPs indeed constitute a specific nosological entity that could be treated without forcefully considering the supposed primary lesion histology and his presumed treatment sensitivity. The suggestion of lack of parallel branching in CUP clonal heterogeneity evolution, if confirmed, could open interesting perspectives in the treatment of these cancers, as could the concept of CUP cell stemness.

As correctly stated by the authors in the discussion section, the main limitation of this work is that a single patient was analyzed, which does not allow to draw firm conclusions and imposes a confirmation of the results in a larger patient sample. With this caveat in mind, the clues from this

thoroughly analysed clinical case are worth of consideration for the readership of the journal.

Here below some minor remarks.

1. The proposed similarity between the tumor phylogenic tree and the galaxy expansion pattern is suggestive but non completely convincing with regard to the branching pattern - this even from a graphical point of view when one see the galaxy image in the supplementary material. Keeping the botanical reference, widely used for this type of representations, the phylogenic tree of this patient's tumor resembles more to an olive tree terminal shoot. The authors should better discuss this issue.
2. In my opinion, the unpublished work of Verginelli et al. should not be cited in the paper if it has not been published. The editor will advice about this point.

Reviewer #2:

The study of Benvenuti et al. aims to shed light on the gene expression pattern and genomic fingerprint of multiple metastasis obtained from a CUP patient and to identify potential therapeutic targets. The results of the manuscript are systematically presented and the paper is written in a clear manner. The authors provide highly interesting data that point to a specific biology that matches the clinical course.

The major claim of the paper states that, according to the evidence presented, the genomic and transcriptomic fingerprints define CUP as unique nosological entity. Compared to molecular signatures of metastatic lesions of known primary origin, the presented case displays a signature different to all other cancer types. In contrast, the sequenced metastases of this case are genetically highly similar suggesting a "galaxy type of evolution".

Major criticism:

This topic has very high relevance and this study may indeed represent a first step towards identifying a novel biology that may indeed consist, as the authors suggest, of a particular degree of stemness in CUP. However, the claim that CUP is a nosological entity is not strongly supported. The authors show that this case is different and special. However, absence of similarity may result from many reasons - in the worst case from laboratory artifacts. Absence of similarity to other cancers in a single case is therefore not enough to define a disease entity. Diagnostically it would be much better to define a positive characteristic, such as the stem cell phenotype.

Additional points

Line 76-78 – there are no figures supporting these claims (e.g. ich, if).

Line 86-87 –it is not clear if the biopsy sample histology presented in Fig 1D comes from the ultrasound guided core biopsy as stated in line 75 or if they were obtained during warm autopsy as stated in lines 82-83 ? I believe that figure with breast metastasis histology is missing.

Line 89- what was the reasoning behind choosing only these metastasis for RNAseq? Please elaborate.

Line 120 – The statement that all metastasis share fraction of mutations of more than 80% seems incorrect, Fig 3A is showing much lower Jaccard index for sample L_08. Samples L_01 and L_04B have lower index as well, around 40%. This should be checked.

Minor corrections:

Throughout the manuscript reference to figures is missing, several typos can be detected or words are omitted. Please check carefully.

Referee #1:

Authors performed a genomic and transcriptomic characterization of several metastases of a case of cancer of unknown primary (CUP). A total of 15 synchronous metastases were retrieved at warm autopsy from 9 different tissues (liver, kidney, breast, lung, axillary lymph node, abdominal subcutaneous tissue, mediastinum, gluteal subcutis and right colic flexure). Whole exome sequencing was performed for the 15 metastases; and RNA sequencing was achieved for 6 metastases (in triplicates).

Comprehensive genomic and transcriptomic data generated in this study provide a valuable raw material to improve our understanding about the mechanistic origin and evolution of CUPs. However, a more exhaustive analysis should be performed to support author's conclusions.

Main comments:

Point 1.1: Transcriptomic analysis (RNAseq): It is extremely surprising that CUP samples segregate completely apart from any other human tissue (TCGA normal, primary and metastatic samples from 33 tissue types). Although a couple of primary ovarian cancer from the group were included to exclude batch effects, additional samples should be included in the analysis to strongly support a distinct expression signature of the studied CUP samples, and robustly discard artifacts.

Answer: To unambiguously show the distinct expression signature of CUP samples, we have now replaced the UMAP analysis with a robust hierarchical clustering analysis based on correlation distance (new Fig. 2) showing that CUP metastases have transcriptional profiles different from the profiles of primary tumors and of metastases from recognizable organs. As noted by the Referee the primary ovarian cancers scrutinized as controls have expression profiles similar to that of ovarian cancers deposited in the TCGA dataset (now highlighted by the purple box, see also point 1.6). Moreover, to further avoid batch effects we have now added a quantile normalization step (Figure EV3A). Methods have been updated accordingly (lines 248-250).

Point 1.2: Please include normal samples from the CUP patient (at least normal liver, kidney, breast and lung) obtained exactly as the analysed CUP samples (warm autopsy). Are they segregating in the expected TCGA normal tissue? Include additional metastasis samples obtained from warm autopsies (if possible, from your same institution).

Answer: Unfortunately, it was impossible to collect normal tissues at the time of warm autopsy, or to take advantage of unrelated cases, in accordance with rules dictated by the Institution Ethical Committee. We think that the new hierarchical clustering presented in new Fig.2 can provide convincing evidence of the uniqueness of CUP transcriptional profiles, making unnecessary further comparison with normal tissues.

Point 1.3: Choose representative individual metastatic samples from TCGA and plot them in Fig. 2. Is the median expression profile preserving the tissue-type identity features?

Answer: We thank the reviewer for her/his suggestion. Starting from median expression profiles of each cancer type and metastases from TCGA, we have now performed a hierarchical clustering analysis to show how median expression profiles preserve the tissue-specific identity features (*i.e.* each metastasis clusters close to its tumor of origin). This new analysis has been now added in Figure EV3B and described in the Manuscript (lines 93-97).

Point 1.4: Considering that authors suggest that their model is consistent with the presence of a common cell of origin with stem-like features... and CUP features suggest a common genetic mechanism that fixes the cells in a "stem" state... it could be interesting to include human stem cells (embryonic and adult stem cells from the literature) in the transcriptomic analysis and plot them in Fig. 2. Are CUP samples showing stem-like traits instead of differentiated-like expression profiles? In addition, gene set enrichment and pathway analysis could be performed using RNA seq data from CUP samples to assess the activation of stem-associated pathways. Moreover, expression of stem-cell markers could be shown. The idea of a common cell of origin with stem-like features is interesting, but some data supporting this hypothesis should be shown. Is the "poorly differentiated" histology consistent with the stem-like traits? Are stem-cell markers expressed in those tissues (IHC)?

Answer: The hypothesis is based on an extensive separate set of experiments described in a manuscript submitted elsewhere and under revision (*Hypermetastatic stem-like cells from 'Cancers of Unknown Primary' model multi-organ dissemination and unveil liability to MEK/MYC inhibition by Verginelli et al.*). The manuscript is attached for the referee eyes only. To avoid duplication the following paragraph was removed:

....This could be explained by properties of CUP stem-like cells. CUP stem-like cells display a keen proliferative autonomy and insensitivity to exogenous factors, which can subtract cells from the selective pressures of the microenvironment shaping the slow and heterogeneous genetic evolution observed in metastases of known origin. Interestingly, in CUPs, unrestrained proliferation, ability to disseminate and home into multiple tissues, and inability to differentiate seem deeply intertwined, and suggest a common genetic mechanism that fixes the cells in a stem state...

Point 1.5.1: Please include methodological and experimental details: Did you perform macrodissection to select tumor cells before RNA extraction?

Answer: Samples were macrodissected to select tumor cells before RNA extraction; after macrodissection percent of tumor cells -assessed independently by 2 pathologists- was above 70% in every single specimen. Methods have been updated accordingly (lines 226-227).

Point 1.5.2: How do you calculate the median expression profile for the TCGA samples?

Answer: TCGA expression data profiles were retrieved as FPKM using the TCGAbiolinks package (Colaprico et al., 2016). The analysis was limited to samples labeled as ‘Primary Solid Tumor’, ‘Primary Blood Derived Cancer - Peripheral Blood’, or ‘Primary Blood Derived Cancer - Bone Marrow’. For each gene and each of the 33 tumor types we extracted the median FPKM across all samples to generate the median expression profile of each tumor type and the profiles transformed into $\log_2(\text{FPKM}+1)$. Methods have been updated accordingly (lines 250-258).

Point 1.6: In addition to the UMAP analysis, please include a clustering analysis. Although UMAP analysis is useful for data visualization and could be used as an efficient preprocessing step, UMAP does not preserve density and linearity. Thus, it is strongly suggested to perform clustering analysis (hierarchical clustering and/or k-means clustering). Please, as previously recommended, include normal tissue samples from CUP patient, other autopsy samples, and stem-cells.

Answer: We agree with the reviewer that UMAP analysis is more useful for data visualization and in majority of cases cannot preserve linearity. For that reason, we have now replaced the UMAP analysis with a hierarchical clustering analysis based on correlation distance to show how CUP samples do not group with their tissue of origin (see point 1.1 and updated Methods; lines 254-258). Concerning the inclusion of normal tissue samples, see point 1.2.

Point 1.7: Genomic analysis (Whole exome seq) and phylogenetic tree reconstruction: One of the most interesting finding of the study is the identification of a single common "stream", sprouting a series of sequential linear "brooks"... suggesting an unusual phylogenetic evolution consistent with the existence of a common ancestor that continues to evolve in a linear fashion... In order to assess the significance of this data, it is necessary to evaluate the context. Thus, it could be useful to include an external "outlier" to measure the significance of the changes.

Answer: As requested, a Jaccard plot including data from WES of three unrelated metastases (CTR_UNR1, CTR_UNR2 and CRT_UNR3) is shown below. The genetic similarity among those controls and metastases of the CUP patient (measured as function of Jaccard index) was zero (white squares). To avoid overloading of the paper we only added this plot in the expanded view (Figure EV4).

Point 1.8: Once revised the transcriptomic and genomic analyses, please rethink the manuscript title, more considering that only one case has been analysed. Although analysis of 15 synchronous metastases from a CUP case is per se very interesting and could improve our understanding about the mechanistic origin and evolution of CUPs and shed lights about the reasons of unusual patterns of dissemination of CUPs, do not forget that is only one case. Before considering CUP as a novel pathological entity & distinct nosological entity based on your genomic analysis, additional cases should be analysed in future studies.

Answer: As suggested by the reviewer, we changed the manuscript title, adding the statement: 'A case report'.

Minor comments:

All minor points have been fixed.

Reviewer #2:

The study of Benvenuti et al. aims to shed light on the gene expression pattern and genomic fingerprint of multiple metastasis obtained from a CUP patient and to identify potential therapeutic targets. The results of the manuscript are systematically presented and the paper is written in a clear manner. The authors provide highly interesting data that point to a specific biology that matches the clinical course.

The major claim of the paper states that, according to the evidence presented, the genomic and transcriptomic fingerprints define CUP as unique nosological entity. Compared to molecular signatures of metastatic lesions of known primary origin, the presented case displays a signature different to all other cancer types.

In contrast, the sequenced metastases of this case are genetically highly similar suggesting a "galaxy type of evolution".

Major criticism:

Point 2.1: This topic has very high relevance and this study may indeed represent a first step towards identifying a novel biology that may indeed consist, as the authors suggest, of a particular degree of stemness in CUP. However, the claim that CUP is a nosological entity is not strongly supported. The authors show that this case is different and special. However, absence of similarity may result from many reasons - in the worst case from laboratory artifacts. Absence of similarity to other cancers in a single case is therefore not enough to define a disease entity. Diagnostically it would be much better to define a positive characteristic, such as the stem cell phenotype.

Answer: As already agreed with Referee #1, we changed the title adding the statement: 'A case report'. Further steps towards the identification of CUP as a novel biology entity are ongoing. Some of them are reported in the enclosed accompanying manuscript (see Point 1.4 raised by Reviewer #1).

Additional points

Line 76-78 – there are no figures supporting these claims (e.g. ich, if).

Answer: Nineteen IHC markers (both positive and negative) used throughout the *ad excludendum* CUP diagnosis are now shown in Figure EV1.

Line 86-87 –it is not clear if the biopsy sample histology presented in Fig 1D comes from the ultrasound guided core biopsy as stated in line 75 or if they were obtained during warm autopsy as stated in lines 82-83 ? I believe that figure with breast metastasis histology is missing.

Answer: Sorry for the flaw. We have now amended text and figures. The histology corresponding to ultrasound guided core biopsy of breast metastasis is now shown in Figure EV1. Fig 1D show specimens collected at warm autopsy. Main text and figure legends have been changed accordingly (line 75).

Line 89- what was the reasoning behind choosing only these metastasis for RNAseq? Please elaborate.

Answer: RNA was extracted from three different regions of each metastasis retrieved at warm autopsy. RNAseq analysis was performed on six metastases (right colic flexure, liver, kidney, mediastinum, breast and lung). The choice was dictated by RNA quality controls. Methods have been updated (line 231-233).

Line 120 – The statement that all metastasis share fraction of mutations of more than 80% seems incorrect, Fig 3A is showing much lower Jaccard index for sample L_08. Samples L_01 and L_04B have lower index as well; around 40%. This should be checked.

Answer: We agree with the reviewer that the statement ‘up to 80%’ may be misleading. Indeed, the genetic concordance of the 15 metastases, measured as a function of Jaccard similarity, ascertains a low degree of inter-metastases heterogeneity. All metastasis (except lesion L_08) display a similarity ranging from 58% to 82% (Fig 3B). The exception of L_08 is due to the scarce number of mutations displayed; nevertheless, out of the 144 somatic SNVs observed, 4 mutations were partially shared and only 4 private. All remaining mutations were fully shared with the other metastases. Metastasis L_01 displays a similarity above 58% with all other metastases with the exception of L_8 (see above) and L_04B displays a similarity above 60%. Jaccard numerical values are reported in the matrix below. The text has been changed accordingly (lines 125-129).

	L_01	L_02	L_03	L_04A	L_04B	L_04C	L_04D	L_07A	L_07B	L_08	L_09	L_10	L_12A	L_12B	L_12C
L_01	1,00	0,61	0,61	0,61	0,60	0,59	0,58	0,61	0,66	0,43	0,61	0,65	0,65	0,61	0,64
L_02	0,61	1,00	0,74	0,73	0,65	0,67	0,72	0,72	0,75	0,37	0,71	0,76	0,73	0,69	0,66
L_03	0,61	0,74	1,00	0,77	0,67	0,72	0,77	0,77	0,79	0,37	0,75	0,82	0,78	0,73	0,69
L_04A	0,61	0,73	0,77	1,00	0,67	0,73	0,74	0,76	0,77	0,39	0,73	0,79	0,77	0,72	0,70
L_04B	0,60	0,65	0,67	0,67	1,00	0,63	0,67	0,65	0,70	0,39	0,66	0,70	0,67	0,66	0,65
L_04C	0,59	0,67	0,72	0,73	0,63	1,00	0,70	0,73	0,72	0,35	0,69	0,74	0,72	0,68	0,65
L_04D	0,58	0,72	0,77	0,74	0,67	0,70	1,00	0,74	0,76	0,35	0,72	0,79	0,75	0,71	0,66
L_07A	0,61	0,72	0,77	0,76	0,65	0,73	0,74	1,00	0,77	0,35	0,74	0,79	0,76	0,71	0,68
L_07B	0,66	0,75	0,79	0,77	0,70	0,72	0,76	0,77	1,00	0,40	0,76	0,82	0,80	0,77	0,73
L_08	0,43	0,37	0,37	0,39	0,39	0,35	0,35	0,35	0,40	1,00	0,38	0,39	0,41	0,42	0,45
L_09	0,61	0,71	0,75	0,73	0,66	0,69	0,72	0,74	0,76	0,38	1,00	0,78	0,76	0,74	0,69
L_10	0,65	0,76	0,82	0,79	0,70	0,74	0,79	0,79	0,82	0,39	0,78	1,00	0,81	0,76	0,73
L_12A	0,65	0,73	0,78	0,77	0,67	0,72	0,75	0,76	0,80	0,41	0,76	0,81	1,00	0,76	0,73
L_12B	0,61	0,69	0,73	0,72	0,66	0,68	0,71	0,71	0,77	0,42	0,74	0,76	0,76	1,00	0,74
L_12C	0,64	0,66	0,69	0,70	0,65	0,65	0,66	0,68	0,73	0,45	0,69	0,73	0,73	0,74	1,00

Minor corrections:

Throughout the manuscript reference to figures is missing, several typos can be detected or words are omitted. Please check carefully.

Answer: We apologize for the typos; we have now carefully revised the text.

Referee #3:

This paper describes the genetic characteristics of several metastatic lesions from a single patient affected by a cancer of unknown origin (CUP), with the aim of getting an insight into the clonal evolution of the disease.

The authors show that the genetic evolution of this case of CUP is "consistent with the existence of a common ancestor that continues to evolve in a linear fashion, releasing over time collateral branches (the different metastases), each of which accrues an independent smaller mutational burden". This is in clear contrast with what we generally know about metastatic cancers.

The materials and methods are appropriate, especially with regard to the selection of the clinical case. The results are clearly explained and compelling.

The findings reported by the authors are quite stunning and could have a significant therapeutic impact. In particular, these results corroborate the hypothesis that CUPs indeed constitute a specific nosological entity that could be treated without forcefully considering the supposed primary lesion histology and his presumed treatment sensitivity. The suggestion of lack of parallel branching in CUP clonal heterogeneity evolution, if confirmed, could open interesting perspectives in the treatment of these cancers, as could the concept of CUP cell stemness.

As correctly stated by the authors in the discussion section, the main limitation of this work is that a single patient was analyzed, which does not allow to draw firm conclusions and imposes a confirmation of the results in a larger patient sample. With this caveat in mind, the clues from this thoroughly analysed clinical case are worth of consideration for the readership of the journal.

Comment: We really thank the reviewer for her/his supporting comments.

Minor remarks

Point 3.1: The proposed similarity between the tumor phylogenetic tree and the galaxy expansion pattern is suggestive but non completely convincing with regard to the branching pattern - this even from a graphical point of view when one see the galaxy image in the supplementary material. Keeping the botanical reference, widely used for this type of representations, the phylogenetic tree of this patient's tumor resembles more to an olive tree terminal shoot. The authors should better discuss this issue.

Answer: As reminded by the reviewer, botanical references are often used to represent phylogenetic trees linking different tumors/metastases (*i.e.* branches, trunk). However, astronomic references as well are used to depict tumor evolution (*i.e.* star, big bang). We agree with the reviewer that the olive tree terminal shoot closely resembles to the phylogenetic tree described. We have now included in text this epitome to illustrate CUP evolution (lines 186-187).

Point 3.2: In my opinion, the unpublished work of Verginelli et al. should not be cited in the paper if it has not been published. The editor will advise about this point.

Answer: As suggested by the reviewer we removed the citation from the manuscript. See point 1.4.

4th May 2020

Dear Prof. Comoglio,

Thank you for the submission of your revised manuscript to EMBO Molecular Medicine. We have now received the enclosed report from the three referees who reviewed the new version of your manuscript. As you will see, they are now supportive of publication, and I am thus pleased to inform you that we will be able to accept your manuscript pending the following final editorial amendments:

1) Title and abstract: please see referee #4's comment on the use of "nosological entity" in the title. After discussion with my colleagues, we would like to suggest: "Cancer of Unknown Primary (CUP): genetic evidence for a novel nosological entity? A case report." Please let me know if this sounds agreeable to you.

2) Main manuscript text:

- Please correct/answer the track changes suggested by our data editors in the main manuscript file (in track changes mode). I will send you this file in the next couple of days.
- Please remove the highlighted text.
- In the Material and Methods section, please indicate the dilution of the antibodies used in the study. Please include a statement that informed consent was obtained from all subjects and that the experiments conformed to the principles set out in the WMA Declaration of Helsinki and the Department of Health and Human Services Belmont Report.

3) Figures:

- Please remove the EV figures from the manuscript and upload them as individual figure files. Fig. EV3 should be uploaded as one single file.
- Tables EV1-3 should be uploaded as separate files, with the file type "Table". Table EV4 should be relabelled "Dataset EV1" and the legend and callout adjusted.
- Please remove the EV table legends from the manuscript and add them directly to the respective files. Legends should only include main figures and EV figures.

4) Thank you for providing a synopsis text and figure. I slightly modified the text to match our format, please let me know if you agree with the following:

Cancer of unknown primary (CUP) is a heterogeneous clinical syndrome characterized by metastases without primary tumor, an elusive biology and lack of effective treatments. We describe the genetic and evolutionary features of 15 synchronous and spatially distinct metastases from a CUP patient.

- CUP metastases were mostly genetically homogeneous, each one accruing an independent smaller mutational burden.
- Reconstruction of phylogenetic relationships among metastases suggested the presence of a common ancestor that linearly and rapidly accumulated mutations, and disseminated a progeny at subsequent stages of its evolution.
- The genomic and transcriptomic traits of CUP metastases prompted recognition of a distinct nosological entity.
- On the clinical ground, the unexpected genetic similarity among different CUP metastases, fosters

a therapeutic strategy aimed at simultaneous eradication of the multiple lesions.

5) As part of the EMBO Publications transparent editorial process initiative (see our Editorial at <http://embomolmed.embopress.org/content/2/9/329>), EMBO Molecular Medicine will publish online a Review Process File (RPF) to accompany accepted manuscripts.

In the event of acceptance, this file will be published in conjunction with your paper and will include the anonymous referee reports, your point-by-point response and all pertinent correspondence relating to the manuscript. Let us know whether you agree with the publication of the RPF and as here, IF YOU WANT TO REMOVE OR NOT ANY FIGURES BEFORE PUBLICATION.

I look forward to reading a new revised version of your manuscript as soon as possible.

Sincerely,

Lise Roth

Lise Roth, Ph.D
Editor
EMBO Molecular Medicine

To submit your manuscript, please follow this link:

Link Not Available

The system will prompt you to fill in your funding and payment information. This will allow Wiley to send you a quote for the article processing charge (APC) in case of acceptance. This quote takes into account any reduction or fee waivers that you may be eligible for. Authors do not need to pay any fees before their manuscript is accepted and transferred to our publisher.

***** Reviewer's comments *****

Referee #1 (Remarks for Author):

The authors have addressed my previous comments and suggestions

Referee #3 (Comments on Novelty/Model System for Author):

I am satisfied with the amendments proposed by the authors.

Referee #3 (Remarks for Author):

None

Referee #2 (Remarks for Author):

The study of Benvenuti et al. aims to shed light on the gene expression pattern and genomic fingerprint of multiple metastasis obtained from a CUP patient and to identify potential therapeutic targets. The results of the manuscript are systematically presented and the paper is written in a clear manner. The authors provide highly interesting data that point to a specific biology that matches the clinical course.

After revision, the authors addressed most points and provided satisfying answers to the questions. My major concern that a single case may not be sufficient to define a nosological entity was addressed by adding "a case report" to the title. I think the paper should be published, because the depth and novelty of the study on this individual case justify it. However, my feeling is that "nosological entity" and "case report" do not go well together. If this is the only case in the world that clusters apart from all other cancers, it will remain an anecdote and will not become a nosological entity. If there are additional examples, I would agree with the authors. However, I do not see any reason why the authors should insist on this term. Even in their discussion they raise it only as a possibility and do not justify the novel nosological entity any further nor do they define the novel entity in any way. Therefore, I recommend to remove the term "nosological entity" from the title. The authors should point it out as a possibility in the abstract. The uniqueness of their case can be highlighted in the title using other words.

The authors performed the requested changes.

Corresponding Author Name: Paolo M. Comoglio
Journal Submitted to: EMBO Molecular Medicine
Manuscript Number: EMM-2019-11756